# Neurogenic Erectile Dysfunction. Where Do We Stand?

**DOI:** 10.3390/medicines8010003

**Published:** 2021-01-07

**Authors:** Charalampos Thomas, Charalampos Konstantinidis

**Affiliations:** 1Urology Department, General Hospital of Corinth, 20131 Corinth, Greece; 2Urology & Neurourology Unit, National Rehabilitation Center, Ilion, 13122 Athens, Greece; konstantinidischaralampos@yahoo.com

**Keywords:** erectile dysfunction (ED), neurogenic, sexual dysfunction (SD), phosphodiesterase type-5 inhibitors (PDE5I), spinal cord injury (SCI), multiple sclerosis (MS)

## Abstract

Erectile Dysfunction (ED) is the persistent inability to attain and maintain an erection sufficient to permit satisfactory sexual performance, causing tremendous effects on both patients and their partners. The pathophysiology of ED remains a labyrinth. The underlying mechanisms of ED may be vasculogenic, neurogenic, anatomical, hormonal, drug-induced and/or psychogenic. Neurogenic ED consists of a large cohort of ED, accounting for about 10% to 19% of all cases. Its diversity does not allow an in-depth clarification of all the underlying mechanisms nor a “one size fits all” therapeutical approach. In this review, we focus on neurogenic causes of ED, trying to elucidate the mechanisms that lie beneath it and how we manage these patients.

## 1. Introduction

Erectile dysfunction (ED) is defined as the persistent inability to attain and maintain an erection sufficient to permit satisfactory sexual performance [1]. Besides the obvious, it has tremendous effects on the patient’s psychosocial health, and it affects not only their quality of life but the lives of their partners [2].

The pathophysiology of ED remains a labyrinth since many pathways can co-exist, thus contributing negatively.

Traditionally ED was divided into three large cohorts, organic, psychogenic and of mixed etiology, but, in most cases, ED is associated with more than one pathophysiological factor, and there is always a psychological component, if not at the beginning, then at least afterwards. Therefore, the most politically correct terms should be primary organic and primary psychogenic.

The pathophysiology of ED may be vasculogenic, neurogenic, anatomical, hormonal, drug-induced and/or psychogenic. 

Neurogenic Sexual dysfunction (NSD) has a prevalence of between 10% and 19% of all causes of erectile dysfunction (ED) [3]. Several reasons may influence ED in patients suffering from neurological diseases, depending on the genesis, which can be central, peripheral, or both. It appears that many neurogenic conditions have overlapping characteristics, for example, patients with diabetes mellitus (DM) may have a vasculogenic and a neurogenic component, therefore the following Table 1 must be used only for diagnostic classification purposes.

Patients experiencing neurogenic ED are a multifaceted group, making management a real challenge. In this narrative review, we focus on neurogenic causes of ED, trying to elucidate the pathophysiological mechanisms that lie beneath this and how we manage these patients, providing a compensatory navigating guide for everyday practice.

## 2. Physiology of Erection 

Sexual response consists of a neuro-vascular event under neuro-hormonal control. Its control is orchestrated by several centers in the brain and the spinal cord. Brain centers are located in the hypothalamus and the midbrain, while two others are located in the spinal cord (sympathetic and parasympathetic autonomous nervous system). The close collaboration of all these centers guarantees a proper sexual response.

Sexual stimuli received via our senses, such as touch, vision, hearing and smell, gather in the temple lobe and, via amygdala, stimulate the upper centers of the hypothalamus. The main involved nuclei are paraventricular nucleus, medial preoptic area, paragiganto-cellular nucleus and locus coeruleus. Those centers are triggered not only by the stimuli mentioned above but also by the recollection of previous sexual memories, experiences and fantasies. The neuronal signal travels to the lower centers of erection, located in the spinal cord: the psychogenic thoracolumbar-sympathetic center (Th11,12–L2,3 level) [4] and the reflexogenic sacral parasympathetic center (S2–S4 level) [5]. Additional afferent signals via the pudental nerve provoke the voluntary contraction of the bulba-cavernosum and cavernosum muscles. Both these muscles, by contracting, compress the crura of the corpa cavernosa over the pubic symphysis, thus increasing pressure inside, leading to a rigid penis. 

The two centers of the spinal cord are under the control of the brain [6]. Depending on the origin of its induction and the erection center mainly involved, two types of erection can be recognized, reflective and psychogenic. On the one hand we have the reflective erection, which is the outcome of somato-aesthetic stimulation and may be independent of sexual arousal. This is facilitated through the reflexogenic, parasympathetic erection center. On the other hand, the psychogenic erection, which predominates in humans, is generated by sexual desire driven by images, fantasies and thoughts related to previous sexual experiences. The psychogenic, sympathetic erection center is mainly responsible for this kind of erection.

Depending on the level of injury (SCI, MS, stoke etc.) and the severity (whether the injury is complete or not), various effects on erectile function can be demonstrated.

## 3. Etiology of Neurogenic ED

### 3.1. Central Nervous System (CNS) Conditions

#### 3.1.1. Spinal Cord Injury (SCI)

SCI is a common disorder, the estimated number of people with SCI living in the United States alone being approximately 294,000 persons. Incomplete tetraplegia is the most frequent neurological category and the frequency of incomplete and complete paraplegia is literally the same [7]. 

Sexual dysfunction is one of the greatest concerns of patients with SCIs, deteriorating their QoL [8,9]. It occurs in almost 80% of them [3] and as mentioned before, sexual dysfunction in these patients has both organic and psychogenic origin. Regarding the level and the extension of injury (incomplete or complete) and the time from injury, different degrees of ED can be observed. Three types of erection can occur after SCI: reflexogenic, psychogenic and mixed. The reflexogenic erection is provoked by direct stimulation of the genitalia and requires an intact parasympathetic center (S2–S4). When the level of injury is above T11 an erection sufficient for penetration, but of small duration, usually occurs [10,11]. Psychogenic erections are independent of direct physical stimulation and are the result of visual or acoustic stimuli, dreams, fantasies and memories [12]. To occur, an intact thoracolumbar sympathetic center (Th11,12–L2,3) is required. These erections are usually characterized by low rigidity and small duration [11,12]. Mixed erections occur when the lesion locates between two centers, below L2 and above S2, producing erections of various duration and rigidity [12,13]. 

Nocturnal erections have also been reported in patients with SCI, occurring during the REM phase of sleep. Quadriplegics seem to have erections harder and longer in duration than paraplegic patients [14] while injuries at the cervical level were associated with better nocturnal erections than those located at the thoracic level [15].

#### 3.1.2. Multiple Sclerosis (MS)

MS is a frequent inflammatory autoimmune disorder of the CNS, affecting about 2.5 million people worldwide. It is the most frequent neurological disorder in young adults and its characteristic is the demyelination of nerve cells in the brain and spinal cord. This damage disrupts the ability of signal transmission, thus resulting in a variety of signs and symptoms, hence the soubriquet “the great imitator”. The prevalence of ED is higher than in the general population and it seems to 70% of men suffering from ED [16]. Because of its diversity, the impact in erectile function depends on the level of the lesion. 

Sexual dysfunction in MS has three distinctive stages: primary, secondary and tertiary sexual dysfunction (SD). Primary SD has to do with the neurological condition of the patient per se, meaning the level and the location of the lesion itself (brain, spinal cord, peripheral nerves). The direct impact on erectile function, the ability of ejaculation, expression of libido and orgasmic reaction depends on the exact extent of the damage. Secondary SD is the result of organic parameters that affect sexual function such as poor bladder and bowel control, muscle weakness, spasticity, immobility, fatigue, cognitive and sensory complaints. Finally, tertiary SD reflects the alterations in sexual function caused by psychological, social and cultural concerns, including low self-esteem, alterations in body image, fear of rejection and how the patients conceive their role in the marriage-relationship [17].

#### 3.1.3. Parkinson’s Disease (PD)

PD is the most frequent movement disorder. It is a chronic neurodegenerative disease characterized by “motor” and “non-motor” symptoms which lead to progressive disability. SD in PD is the most neglected and under-reported aspect of the disease. Very few patients get the attention that is required from their attending physicians [18], and patients do not seek help, mostly out of fear and embarrassment. Dopaminergic pathways responsible for erection and arousal are affected. ED is estimated to affect about 42.6% to 79% [19], while other aspects of sexual function are affected, such as libido, ejaculation and orgasm [20]. 

#### 3.1.4. Multiple System Atrophy (MSA)

MSA is a neurodegenerative disease of undetermined etiology. It is described by parkinsonism and cerebellar, autonomic, urinary and pyramidal dysfunction in various combinations [21,22]. It seems to affect the dopaminergic pathways within the brain, responsible for arousal. ED occurs in the vast majority of these patients and it is usually present before the development of orthostatic hypotension.

#### 3.1.5. Cerebrovascular Accident (CVA/Stroke)

A CVA can occur anywhere in the brain, midbrain, brain-stem and spinal cord, therefore its impact on erectile function depends on the level of the lesion. The estimated prevalence of ED after CVA varies, reported being between 17% and 48% [23]. Disruption of the central network affects erectile function, but medication after the stroke combined with psychological impairment also contribute to ED [24].

### 3.2. Peripheral Neurological Impairment

ED can occur from damage to peripheral nerves, in the cases of pelvic and prostate surgery and diabetes mellitus (DM), but these conditions often have a mixed pathogenetic mechanism, therefore they will be excluded from our discussion.

## 4. Diagnostic Evaluation

The diagnostic algorithm of all types of ED includes medical and sexual history, physical examination, laboratory testing and, in special occasions, advanced work-up, such as nocturnal penile tumescence and rigidity test (NPTR), intra-cavernous injection test, dynamic duplex ultrasound of the penis, arteriography and dynamic infusion cavenosometry or caverno-sonography, and even psychiatric and psychosocial assessment. In fact, care providers use the same algorithm in order to evaluate a neurogenic patient as they would evaluate any other patient suffering from ED, with some remarks to be noted. 

The diagnostic algorithm of sexual disorders as mentioned before in discussing MS has three distinctive stages: primary, secondary and tertiary sexual dysfunction (SD). 

In conclusion, it seems that when dealing with neurogenic ED, it is more necessary than with other types of ED to assess these three levels of SD. 

In Table 2 we summarize the minimal diagnostic evaluation as proposed by the latest EAU Guidelines.

## 5. Management of Neurogenic Erectile Dysfunction

As mentioned before regarding the evaluation of neurogenic ED, the same principles are applied in management. The management algorithm is the same, with some restrictions or footnotes to be taken into consideration.

### 5.1. Oral Pharmacotherapy

#### 5.1.1. Phosphodiesterase Type 5 Inhibitors (PDE5Is)

Up to now, four selective phosphodiesterase 5 inhibitors (PDE5Is) are at our disposal for the treatment of ED in general as first-line treatment [25] (sildenafil, vardenafil, tadalafil and avanafil), but do they work equally on neurogenic ED? Few papers address this issue, and the results are contradictory. Although we have accumulated a great deal of data regarding the efficacy of PDE5Is in neurogenic ED, most of the available research had a small number of participants, lacked head to head comparison and were not double-blind randomized trials. Nevertheless, all have demonstrated efficacy in patients with SCI. What is the proposed dosage? Most patients with SCI are young and they lack the vasculogenic factor, meaning they do not have DM or hypertension, therefore if the lesion isn’t complete, then small dosages of PDE5Is are required [26].

On the contrary, there is a trend towards the usage of higher doses when complete lesions are present, although there is no statistical significance. 

We must be aware that PDE5Is are only effective after an erection has occurred; they do not induce an erection, they sustain or even enforce an erection after it has been established, therefore patients with lesions below T10 have a poor innervation and they might not respond as expected to PDE5Is. In particular, patients with complete lesions at level S2–4 experience the worst failure rates. 

In patients with MS the results are encouraging but can be contradictory. Many data are supporting the efficacy of PDE5Is in such patients, while Safarinejad et al. [27] reported that sildenafil, when compared to placebo, had little effect and therefore cannot be recommended for the routine treatment of ED in MS. The pathophysiological mechanism behind this suggestion was not elucidated.

Patients with PD seem to benefit from the use of PDE5Is [28] and treatment may start with 50 mg, but often an escalation of the dose up to 100 mg might be in order [29].

In conclusion, PDE5Is are highly recommended as first-line treatment in patients with neurogenic ED. In patients with SCI, tadalafil, sildenafil and vardenafil have improved both ED and retrograde ejaculation, along with overall satisfaction as shown from the improvement in IIEF-15 score. Although tadalafil 10 mg seem to be more effective than 50 mg sildenafil 50, all PDE5Is seem to be effective and safe, but we lack high-level studies regarding the safety and side effects of various PDE5Is at any dosage.

#### 5.1.2. Fampridine

Fampiridine is a potassium channel inhibitor used in patients with neurogenic spasticity, which has shown to be beneficial in improving two domains of the IIEF-15 score in both SCI and MS patients [30]. Unfortunately, due to severe adverse effects, a significant discontinuation rate has been reported.

#### 5.1.3. Apomorphine

Apomorphine SL is a D1/D2 dopamine agonist which acts on the paraventricular nucleus of the hypothalamus and might also act on the spinal cord. Patients with an intact thoracolumbar center could achieve psychogenic erections, therefore it might be beneficial for them [31]. The reported rate of success though is low (9.1%), and the main disadvantage of this study was the fact that the population under investigation was heterogenous without a placebo control group. 

### 5.2. Local Therapies

#### 5.2.1. Topical/Intraurethral Alprostadil

Intraurethral administration of alprostadil is an alternative method of delivering this vaso-active agent. At the moment, it is available in two forms, MUSE^TM^ (a medicated pellet) [32] and VITAROS^TM^ (creme) [33], with data coming mostly from the use of the first. Unfortunately, only a few papers address its efficacy in neurogenic ED. Bodner et al. [34] suggested that intraurethral use of 1000 μg of alprostadil (MUSE^TM^) might be effective in inducing erections in men with SCI; however, these erections were less rigid than those achieved using intra-cavernosal administration of the agent. The overall satisfaction rate was low, 3 out of 15 participants. The main adverse effects include urethral pain and hypotension, especially in lesions above T6, since these patients have low blood pressure (90–110 mmHg), therefore a penile constriction ring should be used, in order to prevent systemic absorption leading to a hypotensive episode.

It seems that intraurethral use of alprostadil, in any form, is an alternative treatment for neurogenic patients, less effective though than the traditional intra-cavernous administration, and the most apparent reason of failure in patients with SCI might be the use of intermittent catheterizations, since such a practice alters the epithelium of the urethra to squamous, making the agent less absorbable through it.

#### 5.2.2. Vacuum Devices

Vacuum erection devices can be an optional treatment for ED, showing satisfactory rates of efficiency [35,36]. The vacuum device practically is a tube with an open edge, inside which the penis should be placed. Manually or with electrical assistance, a vacuum is created inside the tube through pumping leading to the creation of negative pressure inside, forcing the blood to fill the corpora cavernosa, thus leading to an erection. A constriction ring is placed firmly surrounding the base of the penis after an erection has been achieved, in order to maintain the erection.

Many researchers have addressed both the efficacy and complications of these devices. Denil et al. studied 20 patients and their partners [37]. After 3 months of use, 93% of men and 83% of women reported erections sufficient for vaginal penetration which lasted for about 18 min. Six months later, however, only 41% of men and 45% of women were satisfied with the use of the vacuum device. The main reported complaint of the users has been the early loss of the rigidity of the erection. 60% of men and 42% of their sexual partners involved in this study reported improvement in their sexual lives. 

Although vacuum devices seem to be a viable alternative treatment for ED, they have their share of adverse effects, including pain, bruising, petechiae or swelling, inability of ejaculation and even gangrene of the penis [38]. Neurogenic patients seem to experience higher complication rates, especially men with SCI, connected with the absence of sensation in the area, so a very tight ring remaining for a long time can cause tissue ischemia and necrosis without any pain or disturbance. Additional precautions should be taken by these patients, in particular avoiding prolonged use and appliance of extremely high negative pressure on the penile shaft.

#### 5.2.3. Intra-Cavernous Injections

In the era of PDE5Is, intra-cavernous injection of vaso-active agents such as alprostadil, papaverine and phentolamine (alone or in combinations of two or three mixes) seems obsolete. However, when PDE5Is fail to deliver a successful response, ICIs become first-line treatment again.

Regretfully, the data we have is from low-quality papers, reflected by the fact that in a recent review and meta-analysis from Chochina et al. [39] only 4 out of 166 articles provided individual data, and given the fact that most papers addressing the use of ICIs in neurogenic patients were published before 1998 when sildenafil was introduced to the practice. 

ICIs are effective in 88% of patients with SCI, with an overall complication rate of 13.3%, with the combination of papaverine with phentolamine being responsible for the higher complication rate of 30%. The most severe adverse effects include ecchymosis (5%), prolonged erection (3–6%), priapism (2–4%), fibrosis (0.5–2%) pain at the site of injection (0.5–2%) and urethrorrhagia (0.1–1.5%). As for dosage, since most neurogenic patients lack a vasculogenic factor, small initial doses are suggested, in order to avoid complications such as priapism. Restrictions are limited since ICIs can be co-administered even with nitrates, whereas PDE5Is are contraindicated, leaving only the dexterity of the patient and his partner.

#### 5.2.4. Low-Intensity Extra-Corporeal Shock Waves Treatment (LI-ESWT)

Ten years have passed since the first published data from Vardi et al. [40] regarding the use of extracorporeal shock waves in the treatment of erectile dysfunction and the data has since accumulated, suggesting the key role they may play in the future.

Even after all these years of extensive research in this field, the underlying mechanism of action of LI-ESWT for ED is still at best unclear. Most of the studies suggest that LI-ESWT provokes the initiation of a cascade of biological events, which includes the up-regulation of vascular endothelial growth factor (VEGF). VEGF is well known for its cell proliferative, anti-inflammatory and anti-fibrotic effects among others, thus leading to angiogenesis, wound healing and tissue regeneration.

How does LI-ESWT work, how does it deliver its therapeutic effect and can it play a significant role in the treatment of neurogenic ED? Although we have promising results from many trials, there is no common protocol, nor there is only one type of machine delivering the shock-wave energy [41]. All the trials exclude patients with pure neurogenic ED. Diabetic patients compose a unique group since there is a mixed underlying mechanism (both vasculogenic and neurogenic, without taking into consideration the negative effect of drugs used for treatment upon erectile function) and therefore should be excluded from this discussion. Patients developing ED after radical prostatectomy were also excluded from the trials, although emerging data from small trials with rather extremely selected patients suggest a beneficial role for LI-ESWT after prostatectomy-cysto-prostatectomy [42,43,44].

Some data are suggesting that LI-ESWT may have a regenerating effect on nerves, especially after nerve injury, which is mediated via VEGF through both a direct neuroprotective effect and by improving the neuronal microenvironment. The treatment significantly reduces cell death and axional damage after Spinal Cord injury (SCI) along with further improvement of locomotor and sensory functions [45], suggesting that LI-ESWT could be added to our armory for the treatment of SCI and consecutively for the definitive treatment of neurogenic ED that follows. However, all the data are experimental in rats and have not yet been verified in human trials, therefore we are far from implementing these methods let alone including the use of LI-ESWT in treatment of neurogenic ED caused by SCI, MS or PD in our everyday practice. 

### 5.3. Regenerative Therapeutic Strategies

In terms of a holistic therapeutic approach when dealing with ED in neurogenic patients, one has to acknowledge that ED is merely one aspect of their misfortune. When discussing the management of neurogenic ED we focus on facilitating an erection, but holistic medicine seeks a way of dealing with the main problem, thus reversing the cause of the neurological condition which troubles the patient. 

In recent years many researchers have focused their efforts on stem-cells and platelet-derived biomaterials contained in platelet-rich plasma (PRP). The reason is obvious since they look promising as the Holy Grail of medicine. Needless to say, these agents have drawn the attention of the researchers studying ED, including that of neurogenic etiology. 

Stem cells can regenerate and restore normal cellular function. Their ability to differentiate into various other cells, along with the secretion of numerous trophic factors, makes them ideal candidates as therapeutic factors in a series of diseases. 

Many types of stem cell have been proposed, such as embryonic stem cells (ESCs), mesenchymal stem cells (MSCs), neural crest stem cells (NCSCs) and endothelial cells. 

ESCs are totipotent cells, meaning they can differentiate into any cell. Their potential use in cavernosal nerve injury was studied [46] but is limited because of ethical concerns.

MSCs can differentiate into many cell types, including adipocytes and neurons. Adipose-derived stem cells have drawn the attention of many researchers because they are easy to obtain by liposuction [47,48].

NCSCs are born during vertebrate embryogenesis within the dorsal margins of the closing neural folds. They have the ability to differentiate into various cell types, including peripheral neurons and Schwann cells (glia) [49]. When transplanted into the cavernosum of adult rats they regenerated endothelial and smooth muscle cells [50].

Endothelial cells were used to promote endothelial repair in diabetic rats via enhancement of VEGF165 expression and neovascularization [51].

Stem-cell therapy could be very promising in treating neurogenic ED, although we are far from using it in everyday practice [52].

### 5.4. Penile Prostheses

The surgical implantation of a penile prosthesis (PPI) is the last and definitive option in terms of treating ED and it should be considered when all other means of treatment have either failed or are unacceptable to the patient. Two kinds of implants are used, the semi-rigid (malleable) and the inflatable (two or three pieces). Both are not free of complications, the main being mechanical failure and infection. Even if implemented by highly experienced surgeons and in special centers, and despite the use of appropriate antibiotic prophylaxis against Gram-positive and gram-negative bacteria or even the use of specially designed devices, such as the antibiotic-impregnated prosthesis (AMS Inhibizone™) or hydrophilic-coated prosthesis (Coloplast Titan™), the rate of infection is still up to 2–3% in low-risk patients. 

When applied to neurogenic patients, apart from patients recovering from radical prostatectomy, the vast majority are referred to patients with SCI or DM. Unfortunately, we have little data published in the literature concerning the clinical outcomes of PPI surgery in neurogenic ED. 

In general, diabetics are at higher risk of developing an infection after a penile prosthesis implementation, since the underlying hyperglycemic environment that describes these patients favors immune dysfunction, meaning poor response in terms of wound healing. Diabetics with ED are a special group of patients, who require additional care when penile prosthesis implantation is to be considered. Besides the role of perioperative antibiotic prophylaxis and the use of specially coated devices, strict monitoring and glycemic control is in order, before and long after the implantation.

Regarding purely neurogenic patients, such as patients suffering from SCI, special considerations should be taken under advice. Compared to non-neurogenic men, the complication rates are higher, classifying them as unsuitable for penile prosthesis implantation as shown by Kim et al. [53] who reported a total complication rate of 16.7%.

The reasons are many. Many patients with SCI lack dexterity, thus making the choice of a malleable prosthesis seem more appropriate for them. Many lack sensibility, bear a suprapubic or condom catheter or use intermittent catheterizations leading to bacteriuria, suffer from decubitus ulcers and recurrent UTIs often associated with neurogenic bladder, and, last but not least, their disease is progressive, meaning we cannot predict the outcome. On the other hand, proper implantation of PPI offers the opportunity to address simultaneously other aspects of everyday concern to these patients besides ED, such as urinary incontinence, making the application of urinal condoms feasible [54].

In conclusion, even with modern achievements and scientific innovations in PPI technology, special care should be given both in selecting and treating these patients and they should be counseled regarding the higher rates of complication and revision needed [55]. 

In Figure 1 we summarize the management algorithm for ED, as proposed by EAU, with remarks, especially for neurogenic ED.

## 6. Conclusions

Neurogenic ED is characterized by a tremendous diversity since it has many etiological factors. This diversity makes it impossible to fully understand the underlying mechanisms of its pathophysiology, let alone to design a “one fits all” method of treatment. Several studies are needed, not only to elucidate the reason why ED is established in several neurological conditions, but also to identify the pathways that are involved in different conditions, how they intersect and, finally, how we can provide a “tailor-made” therapeutic approach to every one of our patients, trying to meet their needs successfully.

## Figures and Tables

**Figure 1 medicines-08-00003-f001:**
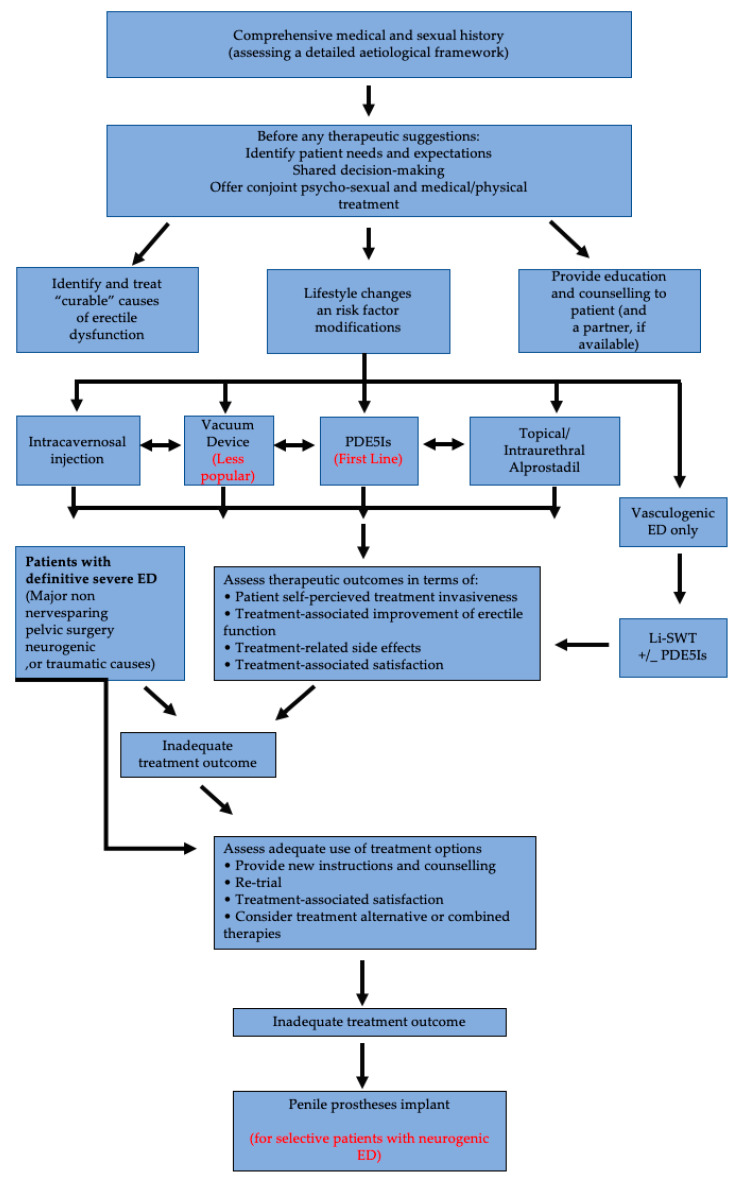
Management algorithm for ED. Adapted from EAU Guidelines on Sexual and Reproductive Health 2020 [1].

**Table 1 medicines-08-00003-t001:** Etiological factors of neurogenic erectile dysfunction (ED).

Neurogenic
**Central** **Causes**
Degenerative disorders (e.g., multiple sclerosis, Parkinson’s disease, multiple atrophy, etc.)
Spinal cord trauma or diseases
Stroke
Central nervous system tumours
**Peripheral Causes**
Type 1 and 2 diabetes mellitus
Chronic renal failure; chronic liver failure
Polyneuropathy
Surgery (major surgery of pelvis/retroperitoneum) or radiotherapy (pelvis or retroperitoneum)
Surgery of the urethra (urethral stricture, urethroplasty, etc.)

Adapted from European Urology Association (EAU) Guidelines on Sexual and Reproductive Health 2020 [1].

**Table 2 medicines-08-00003-t002:** Minimal diagnostic evaluation in patients with ED (including neurogenic ED). Adapted from EAU Guidelines on Sexual and Reproductive Health 2020 [1].

Patient with ED (Self-Reported)					
Medical and Psychosexual History (Use of Validated Instruments, e.g., IIEF) History of Neurological Condition					
Identify other sexual problems, (not ED)	Identify common causes of ED (neurological conditions should be recognized here)			Identify reversible risk factors for ED	Assess psychosocial status
Focused Physical Examination					
Penile deformities	Prostatic disease	Signs of hypogonadism		Cardiovascular and neurological status	
	Check sensation of perinatal area and integrity of reflective arc sacral parasympathetic centre (S2–S4) (Bulbocavernosus and anal reflexes)				
Laboratory Tests					
Glucose-lipid profile (if not assessed in the last 12 months)			Total testosterone (morning sample) if indicated, bio-available or free testosterone		

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
