# Peer review of "Neurogenic Erectile Dysfunction. Where Do We Stand?"

_medicines, 2021, doi:10.3390/medicines8010003_

Round 1
Reviewer 1 Report
The information provided in this paper gives a comprehensive overview of neurogenic ED. However, it is unclear from the introduction what the aims of this paper are. Is it a literature review? Is it a narrative review? Some further clarification during the introduction would help the reader to ascertain what information they are going to be getting. For example, this literature review is divided into a number of sections highlighting the physiology, diagnostic evaluation and management. Similarly, it would be beneficial to provide further information about why this review is needed and how it contributes to the wider knowledge base. How is it transferrable to practice and to whom is it most important?
Some minor errors in Line 55 (should read "previous" and "fantasies". Line 68 needs to be re-worded as the sentence is incomplete.
Author Response
Dear reviewer, thank you for your kind words and for your insight. As you can see from the revised draft we have taken under consideration your remarks. Obviously, it is a narrative review. Verbal and grammatical mistakes have been corrected, and most important we believe that in the introduction we make clear our purpose: to provide a short navigational guide for physicians who deal with such patients, making their everyday practice a little bit easier.
Reviewer 2 Report
Dear Authors
I would like to commend for a manuscript well written. It certainly provides more comprehensive evaluation and treatment for erectile dysfunction in the neurogenic related disease in men. An area that is certainly challenging to practicing urologist.
I would like to follow up with below comments
- Minor grammar and spelling in manuscript please revise when see appropriate
- Please provide some illustruation and tables to make reading your manuscript more attractive and informative
- Provide more details as why the neurogenic patients with erectile dysfunction are at high-risk complications if receiving PDE5i, ICI or penile implant? compared with general population. What is unique and critical to assess prior recommending treatment for ED? This is crtiically important for urologists
Author Response
Dear reviewer, thank you for your kind words and for your insight. As you can see from the revised draft we have taken under consideration your remarks.
Verbal and grammatical mistakes have been corrected.
With respect, regarding your remarks about why patients with neurogenic ED are at high risk of experiencing complications we believe that the revised manuscript is more elucidating (please see paragraphs 5.1.1., 5.2.3. and 5.4).
Tables and figures have been added as recommended, hoping that the paper is more appealing to readers.
Reviewer 3 Report
the manuscript does not bring news in the andrological field. there are already revisions of this kind in the literature.
Author Response
Dear reviewer, with respect to your remarks, the aim of this manuscript is to provide a comprehensive understanding of neurogenic erectile dysfunction, thus offering a useful tool in everyday clinical practice.